# Evaluating the efficacy of various traps in catching tsetse flies at Nech Sar and Maze National Parks, Southwestern Ethiopia: An Implication for *Trypanosoma* Vector Control

**Netsanet Asfaw**[1,2]*, **Berhanu Hiruy**[2], **Netsanet Worku**[3], **Fekadu Massebo**[2]*

**1** National Tsetse Fly and Trypanasomiasis Control and Eradication Institute, Arba Minch Tsetse Fly and Trypanasomiasis Control and Investigation Center, Arba Minch, Ethiopia, **2** Department of Biology, College of Natural Sciences, Arba Minch University, Arba Minch, Ethiopia, **3** Institute of Public Health, College of Medicine and Health Sciences, University of Gondar, Gondar, Ethiopia

* netsataye2007@gmail.com (NA); massebofekadu@gmail.com (FM)

## Abstract

Tsetse flies are the vector of protozoan parasite of the genus *Trypanosoma*, the causative agent of human African sleeping sickness and animal trypanosomiasis. Traps such as Nguruman (NGU), biconical and sticky traps are in use for tsetse flies sampling and monitoring. However, there is no evidence regarding their comparative efficiency in catching flies using olfactory cues. Therefore, the present study aimed to evaluate the efficiency of different types of traps in catching tsetse flies at Nech Sar and Maze National Parks, Southwestern Ethiopia. The study was done for six consecutive months from February to July 2019. Briefly, a 3×4 Latin square design was performed, and tsetse flies were collected for three days each month in four different vegetation types, including wood grassland, bush land, forest, and riverine forest. To avoid trapping position bias, rotation of traps has been done every day. Almost all (99.5%) of the flies were *Glossina pallidipes* and the remaining were *G. fuscipes*. The latter were present only at Maze national park. NGU traps were the most efficient type with 12.1 flies/trap/day at Nech Sar National Park and it was 2.2 flies/trap/day at Maze National Park followed by biconical and sticky traps. The number of tsetse flies collected by biconical trap was three-fold lower than NGU trap, and it was four-fold lower in sticky trap than NGU trap in both Nech Sar and Maze National Parks. A substantial number (41%) of *G. pallidipes* were collected from woody grassland (WGL). In conclusion, *G. pallidipes* monitoring and evaluation activities could consider NGU trap model as it performed better in most vegetation types in the region.

## Author summary

African Animal Trypanosomiasis (AAT) or Nagana and Human African Trypanosomiasis (HAT) agents, are transmitted by a blood feeding vector called tsetse flies. In Animals, the disease is arguably the main constraint to an integrated livestock and crop production in

**Funding:** This work was financially supported by the Norwegian Programme for Capacity

Development in Higher Education and Research for Development-Arba Minch University (ETH-13/0025). The grant recipient (FM) received no salary from the project. The funders had no role in study design, data collection and analysis, decision to publish, or preparation of the manuscript.

**Competing interests:** The authors declare that there is no conflict of interest.

sub-Saharan Africa, including Ethiopia. In humans, the disease is highly neglected. Vector control is considered to be one of the strategies to control the AAT and HAT. These interventions, however, need sampling and monitoring of vector distribution in disease endemic areas. This will be done using different tsetse traps. In Ethiopia, especially in southern and south-western regions, NGU (Nguruman), Biconical and Sticky traps are commonly used, but no evidences on the effectiveness of these traps in catching flies have been generated. Therefore, the measurement of traps efficiency and identification of the most effective one based on the type of vegetation in Nech Sar and Maze National Parks was the aim of this study. Our findings indicated that the NGU trap was the most effective model and the best-identified trap in most vegetation types in the region. This will help in monitoring the impact of interventions and for effective control of trypanosomiasis.

## Introduction

Tsetse flies (Diptera: *Glossinidae*) are responsible for transmission of the protozoan parasites of the genus *Trypanosoma*, the pathogenic agents of Human African Trypanosomiasis (sleeping sickness) and Africa Animal Trypanosomiasis (nagana) [1, 2]. There are 33 species and sub-species of tsetse flies in Africa, infesting over 10 million km$^2$ areas in 37 Africa countries [2]. Both male and female tsetse flies are important vectors of human and animal trypanosomiasis. The parasites are injected into the hosts when an infected tsetse fly bites the susceptible hosts to take a blood meal [3]. Animal trypanosomiasis results in loss of weight, loss of productivity, reduced milk yield and abortion in the infected animal [2].

Since early times, there are several tsetse fly control methods, namely bush clearing, game animal elimination, hand catching, ground and aerial spraying [4]. But, these techniques had not been considered to be effective due to their limited effect on the target vector, their high cost and environmental consequences [3, 4]. Consequently, several other techniques have been developed and widely used in almost all tsetse infested African countries, including Ethiopia. These methods include stationary targets, ground and aerial spraying of insecticides, mobile targets, sterile insect technique (SIT) and integrated vector management strategies [5, 6].

There are also several types of traps for monitoring the impact of interventions and sampling tsetse flies. However, most traps are developed in West and Southern Africa for surveillance of riverine and savannah tsetse species. Moreover, the efficacy of traps in capturing tsetse fly varies from species to species and from location to location. Therefore, there is a need to modify the design of traps to enhance the efficacy of traps based on the species commonly found in a particular target area [7, 8]. Usually, traps function through visual stimuli. In the field, however, the visual stimuli can be greatly obstructed by vegetation, particularly in forest habitats. In such cases, attraction of flies to traps is enhanced with odor attractants. At least three groups of natural odors (urine, breath and skin secretions) have been identified so far from the host animals as bait [5, 9]. In Ethiopia, Nguruman (Ngu trap) and biconical traps are widely in use. In addition, sticky traps are in use for trapping flies. However, little is known about the relative effectiveness of these traps in monitoring and assessing tsetse fly control interventions using olfactory cues (baits) in various types of vegetation.

In the Southern Rift Valley of Ethiopia, there is an on-going program aiming to eradicate tsetse and trypanosomiasis. The program has been undertaken for about 20 years. These strategies include deployment of odour baited and mobile targets, ground and aerial spraying and chemical impregnated targets to suppress the tsetse population. Most recently, the male sterile

technique has been widely deployed once the tsetse flies population was suppressed [10]. At the final stages of vector elimination, there should be a very sensitive trapping methods for surveillance and monitoring and ensure tsetse elimination [1, 10]. Therefore, identifying the efficient traps type has a paramount importance for routine follow-up and monitoring of the tsetse species dynamics that can aid in decision-making to undertake control interventions in an integrated fashion. Accordingly, the purpose of this study was to compare the trapping efficiency of three types of traps in different vegetation types using cattle urine baits. The study also assessed the current abundance and infestation of tsetse flies and monthly variation in both national parks studied.

## Materials and methods

### Ethical considerations

Permission to conduct the study in the National Parks was obtained from Nech Sar and Maze National Parks administrators. The objective of the study was discussed. Training was given to data collectors on collection techniques and park rules and regulations.

### Study areas description

This study was carried out in Nech Sar and Maze National Parks in Southwest Ethiopia (Fig 1). The two parks are located in the Rift Valley of the Southwest Ethiopia in Gamo zone. The total area of Nech Sar National Park is 514 km$^2$ and the Maze National Park covers about 210 km$^2$. The elevation of the Nech Sar National Park ranges between 1108 and 1650 meters above sea level (masl). Its annual temperature ranges between 20˚C and 34˚C. The park forms the riverine forest vegetation type along the Kulfo River that flows to the Lake Chamo. This may create a good habitat for tsetse flies. There are also other habitats like savannah or wood grass land, bush land and forest [11].

Maze National Park, get its name from the Maze River which crosses the park, has an elevation ranging between 900 and 1200 masl. The park is mainly covered by savannah grassland with scattered deciduous broad-leaved trees as well as riverine forest along the main watercourses. The riverine vegetation along the banks of the Maze River is an ideal habitat for tsetse fly.

A number of wild animals including Zebras, Gazelles, Hogs, Hippopotamus, Lions, Leopards, Antelopes, Buffaloes, Swayne's Hartebeests and others are living in the Nech Sar and Maze National Parks and serving as a source of blood meals for tsetse flies [12].

### Study design and period

A Latin-Square design was employed to evaluate the efficacy of the three trap types, namely NGU, Biconical and Sticky traps in four vegetation types (wood grassland, bush land, forest and riverine forest vegetation). The traps are made from cotton. In brief, NGU trap has two sides blue. The black target base is attached half way along the base of the two sides and its top is fixed to the upper rear corner [13]. Biconical trap consists of two cones each 80 cm wide, an upper cone 73 cm high and a lower cone 60 cm high, joined at their widest point. The blue lower cone has four entrances, approximately 30 cm high and 20 cm wide. Vertically dividing the inside of the trap is a black cruciform [14]. A 12 mm hole on its apex of NGU and biconical traps admits flies to the cage. Sticky trap comprised of blue and black panels each 0.25 × 0.25 m in size and thus making it 0.25 × 0.50 m in dimension [14]. The board with the fastened targets was then covered with a transparent sticky film and it smears adhesive TEMO-O-CID Color on the film (GR 570 22 Thessaloniki, Greece).

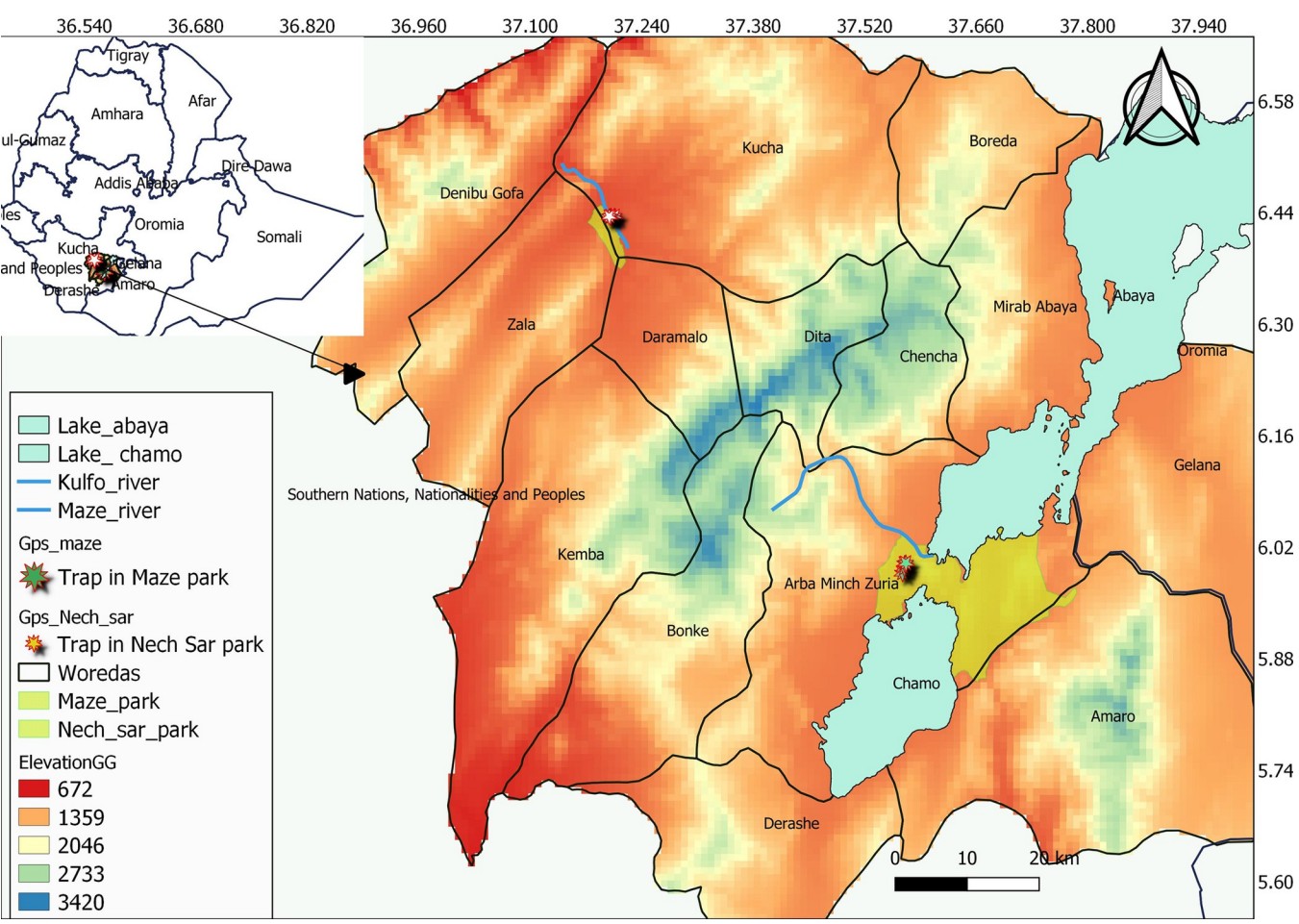

**Fig 1. Map of the study area in Gamo zone, Southwest Ethiopia.** Administrative boundaries, USGS Earth Explorer: (https://earthexplorer.usgs.gov), Roads, Water (River, lakes))–from DIVA-GIS (https://www.diva-gis.org/gdata), Central Stastical Agency and Regional of Finance and Economies Development).

The tsetse collection was done for six consecutive months between February and July, 2019. The number of Latin square replicates in each vegetation type was three per study site/month. In brief, 9 traps were deployed per month per survey site per vegetation. A total of 36 traps were deployed per site/month. Over the six consecutive months, 216 traps were set at each survey site. There were 432 traps deployed at both sites. The initial trapping site for each vegetation type was selected at random. Next, traps were rotated clockwise every day for three days each month in order to minimize bias due to trap placement and day effect. The distance between the traps was placed in a 200 m long equilateral triangle in open vegetation like WGL and BUL, while 100 to 150 m side lengths in closed or dense vegetations such as forest and riverine forests.

## Tsetse collection and species identification

The tsetse fly collection was done using different trap models baited by fermented cattle urine. All information, including date of deployment, date of collection, code of collectors, fly species, season of data collection, type of vegetation and traps, and GPS coordinate was recorded on a standardized data sheet.

Tsetse flies were sorted by species, sex, vegetation type, trap model, GPS position, and month of collection. The identification of the tsetse species was based on morphological characteristics using a hand-held lens, as indicated in the tsetse classification keys [15, 16]. Morphological features such as the antenna, antennal fringe, colour of underside of thecal bulb, and thorax, median bristles on scutellum, wing spot, colour and segments of leg, colour of abdomen and type of superior claspers were considered [15, 16].

## Outcome variables

The primary outcome variable was the number of tsetse flies trapped in each trap and vegetation type. The composition of the tsetse species and its monthly distribution by vegetation type were also evaluated as secondary variables.

## Data management and analysis

The data was entered in Microsoft excel and transferred to the statistical package for social sciences (SPSS) software version 20.0 for analysis. The non-parametric statistical tests, Kruskal-Wallis and Mann Whitney U Test, were applied to test the mean variation between traps and vegetation types. The number of tsetse flies collected served as a dependent variable. The collection site, trap type, tsetse species, vegetation type and month of collection were the independent variables. The total number of tsetse flies and the number of traps in each vegetation type and research site were used to calculate the mean number and density of tsetse flies. Analysis of results was considered statistically significant when the *P*-value was $< 0.05$ at a 95% confidence interval.

## Results

### Tsetse fly species

*Glossina pallidipes* was the dominant (1592/1599; 99.6%) species in both parks. A few (7/1599; 0.4%) *G. fuscipes* were collected at Maze National Park in the iverine forest vegetation using sticky traps. NGU and biconical traps were not successful in catching *G. fuscipes*. None of the *G. fuscipes* were caught in Nech Sar National Park. Eight three per cent (1317/1599) of *G. pallidipes* were collected from Nech Sar National Park, while the remaining 17.3 per cent (282/1599) were collected from Maze National Park.

### Monthly variation of G. pallidipes

Over 60% of *G. pallidipes* were captured in the first three months (February, March and April) and the remaining 40% were captured in May, June and July (Fig 2).

### Overall density of G. pallidipes and traps efficacy

The overall mean number of G. pallidipes was 7.8/NUG trap, and it was 2.2/biconical and 1.7/sticky trap. The average rank of NUG trap was 275.6 and it was 199.3 for biconical and 174.5 for sticky trap. There was statistically significant variation between traps in catching G. pallidipes (Kruskal-Wallis Test $\chi^2 = 54$; DF = 2; p $< 0.001$). Statistically, more G. pallidipes were collected by NGU trap than biconical (Mann-Whitney U test $\chi^2 = 76.3$; p $< 0.001$) and sticky trap (Mann-Whitney U test $\chi^2 = 101.1$; p $< 0.00$). No statistically significant difference was observed in the number of G. pallidipes collected by biconical and sticky traps (Mann-Whitney U test $\chi^2 = 24.8$; p = 0.25).

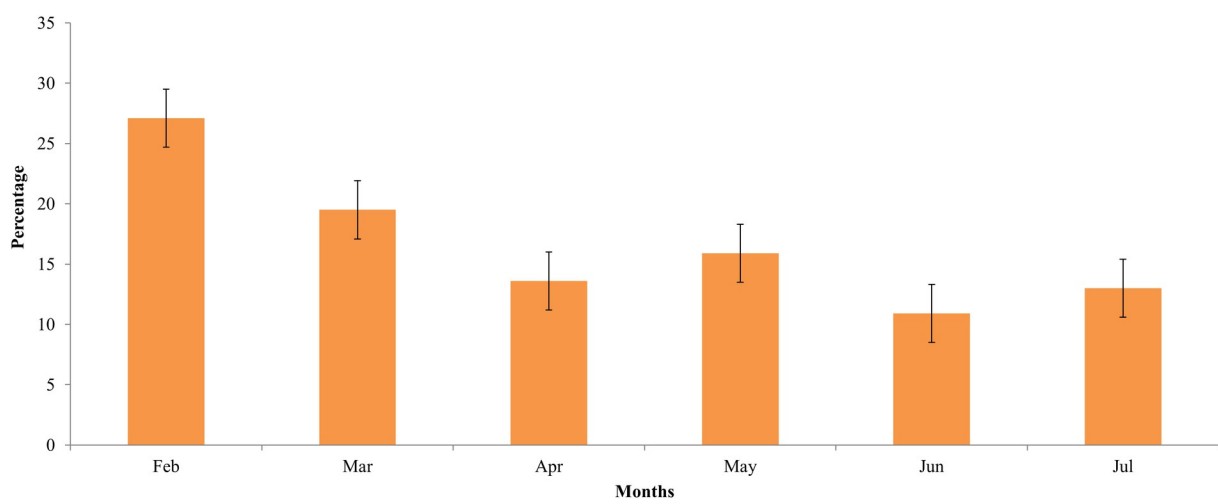

**Fig 2. The overall monthly distribution of tsetse flies in the study area.**

## Density of *G. pallidipes* and traps efficacy in the two parks

The mean number of *G. pallidipes* in Nech Sar National Park was 6.1 flies/trap/day, while it was 1.3 flies/trap/day at Maze National Park (Table 1). The Mann-Whitney U Test mean rank was 269.9 in Nech Sar National Park and this was significantly higher than the mean rank value of 163.1 in Maze National Park ($p < 0.001$). There was variation in traps performance in the two National Parks. NUG trap had better performance in catching *G. pallidipes* than other traps. The mean number of *G. pallidipes* per NUG trap was 12.1/trap in Nech Sar National Park. The same trap type showed better performance in the Maze National Park with mean number of *G. pallidipes* of 2.2/trap. This number of *G. pallidipes* has decreased nearly by three-folds when using biconical traps at Nech Sar and Maze National Parks and four folds when using sticky traps at Nech Sar National Park.

The average rank value (Kruskal-Wallis Test pairwise comparison) of NGU trap in Nech Sar National Park was 150.4, which was significantly higher than the average rank value of biconical (95.5) and sticky trap (79.6). NGU trap also showed better performance in Maze National Park, with the average rank value of 144.2, which was significantly higher than the average rank value of biconical trap (95.2) and sticky trap (89.2). The variation between biconical and sticky traps in catching *G. pallidipes* was not statistically significant in the two parks (Table 2).

## Traps efficacy in different vegetation types

Of the 1599 tsetse flies collected, 62% (990/1599) were caught by NGU traps, 343 (22.4%) by Biconical traps and the rest 266 (16.6%) were trapped by Sticky traps (Table 3). NGU trap was

**Table 1. Mean density of tsetse flies in Nech Sar and Maze National Parks, Southwest Ethiopia.**

| Study sites | NGU, n (Fly/ trap/day) | Biconical, n (Fly/ trap/day) | Sticky, n (Fly/trap /day | Total, n (Fly /trap/day) |
|---|---|---|---|---|
| Nech Sar | 814 (12.1) | 284 (3.5) | 219 (2.7) | 1317 (6.1) |
| Maze | 176 (2.2) | 59 (0.85) | 47 (0.81) | 282 (1.3) |
| Total | 990 | 343 | 266 | 1599 |

n = number of fly

**Table 2. Efficacy of traps in in catching *G. pallidipes* in Nech Sar and Maze National Parks, Southwest Ethiopia.**

| | Tsetse collection sites | | | | | | | |
| --- | --- | --- | --- | --- | --- | --- | --- | --- |
| | Nech Sar National Park | | | | Maze National Park | | | |
| Trap models | $\chi^2$ | Standard error | Standard test statistics | P value | $\chi^2$ | Standard error | Standard test statistics | P value |
| Sticky versus Biconical | 15.9 | 10.3 | 1.5 | 0.36 | 6.0 | 9.7 | 0.62 | 1.00 |
| Sticky versus NGU | 70.8 | 10.3 | 6.8 | <0.001 | 52.1 | 9.7 | 5.4 | <0.001 |
| Biconical versus NGU | 54.8 | 10.3 | 5.3 | <0.001 | 46.0 | 9.7 | 4.7 | <0.001 |

most efficient in all vegetation types, except in riverine forest. Of the 656 tsetse flies collected from WGL, 70.6% (463/656) were collected by NGU trap, followed by Biconical (21.2%). Similarly, 63.5% of the tsetse from BUL and 66% from Forest were collected by NGU trap. Sticky traps performed better in riverine forests (58.3%) compared to NGU traps (23%) and biconical traps (18.6%).

The average rank value (Kruskal-Wasllis Test pairwise comparison) of NGU trap was 99.6 in WGL, followed by 80.6 in BUL, and 75.0 in forest. The lowest average rank value (34.8) was documented in riverine forest. The performance of the NGU trap was significantly higher in the WGL, BUL and forest than in the Riverine forest (Table 4). With regard to biconical trap, its average rank value was maximum (89.1) in WGL, followed by 74.4 in forest, and 69.7 in BUL. The lowest average rank value (56.7) was recorded in the riverine forest. The biconical trap showed a better performance in WGL compared to the riverine forest, and not in the other vegetation type. Unlike the other trap types, the highest average rank value of sticky trap (98.9) was recorded in the riverine forest, followed by BUL (66.4) and WGL (65.4). The lowest was recorded in forest with the average rank value of 59.3. The performance of sticky trap was significantly higher in the riverine forest than in the BUL, forest and WGL.

## Discussion

The results of our study indicate that the NGU trap was better at catching *G. pallidipes* in all vegetation types than the biconical and sticky traps. Sticky trap was more suitable for riverine forests. Our findings are consistent with a study carried out in Tanzania [17] in which the NGU trap was effective in catching *G. pallidipes*. The same trap was more efficient at trapping *G. morsitans submorsitans* in the upper valley of the Didessa River in Ethiopia [18]. NGU trap was reported as the most efficient and the suitability of the NGU trap for community interventions against tsetse fly was reported in Kenya [19]. In another study in Uganda from three trap models (Monoscreen, Pyramidal and Biconical trap), biconical trap was the second important trap next to Monoscreen

**Table 3. Overall number of flies collected from different vegetation types using the three trap types in Nech Sar and Maze National Parks, Southwest Ethiopia.**

| Vegetation types | Study sites | Trap models | | | |
| --- | --- | --- | --- | --- | --- |
| | | NGU (%) | Biconical (%) | Sticky (%) | Total (%) |
| WGL | Nech Sar | 361(69.4) | 108(20.8) | 51(9.8) | 520 |
| | Maze | 102(75) | 31(22.8) | 3(2.2) | 136 |
| BUL | Nech Sar | 242(64.4) | 82(21.8) | 52(13.8) | 376 |
| | Maze | 21(55.3) | 9(23.7) | 8(21.0) | 38 |
| Forest | Nech Sar | 189(66.5) | 64(22.5) | 31(10.9) | 284 |
| | Maze | 29(63.0) | 12(26.1) | 5(10.9) | 46 |
| Riverine | Nech Sar | 22(16.1) | 30(21.9) | 85(62) | 137 |
| | Maze | 24(38.7) | 7(11.3) | 31(50.0) | 62 |
| | Grand total | 990 (62) | 343 (22.4) | 266 (16.6) | 1599 |

**Table 4. Comparison of efficacy of different traps in catching *G. pallidipes* in different vegetation types, southwest Ethiopia.**

| Vegetation type | NUG trap | | | | Biconical trap | | | | Sticky trap | | | |
|---|---|---|---|---|---|---|---|---|---|---|---|---|
| | $\chi^2$ | Standard error | Standard test statistics | p value | $\chi^2$ | Standard error | Standard test statistics | p value | $\chi^2$ | Standard error | Standard test statistics | p value |
| Riverine forest versus forest | 40.2 | 9.7 | 4.1 | <0.001 | 17.6 | 9.4 | 1.8 | 0.37 | 1006 | 84.8 | 4.2 | <0.001 |
| Riverine forest versus BUL | 45.8 | 9.7 | 4.7 | <0.001 | 13.0 | 9.4 | 1.4 | 1.00 | 942.0 | 85.6 | 3.4 | 0.002 |
| Riverine forest versus WGL | 64.7 | 9.7 | 6.6 | <0.001 | 32.3 | 9.4 | 3.4 | 0.004 | 947.0 | 85.5 | 3.5 | 0.001 |
| Forest versus BUL | 5.6 | 9.7 | 0.57 | 1.00 | -4.6 | 9.4 | -0.48 | 1.00 | 582.5 | 76.2 | 0.86 | 1.00 |
| Forest versus WGL | 24.5 | 9.7 | 2.5 | 0.07 | 14.7 | 9.4 | 1.5 | 0.73 | 595.5 | 76.2 | 0.68 | 1.00 |
| BUL versus WGL | 18.9 | 9.7 | 1.9 | 0.31 | 19.3 | 9.4 | 2.0 | 0.25 | 656.5 | 77.8 | 0.11 | 1.00 |

BUL = Bush land; WGL = Woody Grassland

for catching *G. fuscipes* [20] but in Burkina Faso Biconical trap was effective for *G. tachinoides* and *G. palpalis gambiensis* [21]. Though these two studies did not use sticky trap, biconical trap was effective for catching riverine tsetse species. In the current study, *G. pallidipes* was the most distributed species unlike those studies in Burkina and Uganda. Besides, NGU trap was not among the list of traps applied in these two previous studies.

Majority of *G. pallidipes* flies were collected in WGL and the lowest was in the riverine forest of the two sites. The current result is consistent with earlier studies in Nech Sar National Park [12, 22]. Similar results were reported by other studies [18, 22]. This could probably be due to the fact that WGL are suitable for the reproduction of *G. pallidipes* and their propagation, as WGL provides shade and humid habitats [23]. To that end, Nech Sar Park, where over 82% of tsetse flies were caught, has an annual temperature ranging from 20˚C to 34˚C which corresponds to the range of temperature required for the general survival and breeding of tsetse flies and for the third instar larva to fully grow or change into a pupa stage [23]. This could serve as practical evidence of the importance of climate and vegetation in determining the distribution of tsetse flies [24].

Our study found that the density of *G. pallidipes* was significantly higher in Nech Sar National Park compared to Maze National Park. On the other hand, the number of *G. pallidipes* collected in both parks was low compared with previous studies [12, 22]. This may provide a good indicator of the effectiveness of vector suppression activities in both parks. For example, four years ago, the density of *G. pallidipes* in Nech Sar National Park was 47.8 flies/trap/day, nearly 8 times greater than the current density [12, 22]. Tsetse fly suppression activities such as ground spaying, deployment of chemically impregnated targets, and movable/animal targets [25,26] have been implemented to control the tsetse population in both parks. These vector control activities were undertaken at the borders of Nech Sar National Park but these suppression activities were on-going in and around the Maze National Park. This could be a major reason for the low abundance of tsetse flies in Maze National Park relative to Nech Sar National Park. Making conclusions regarding the annual density of tsetse flies in the research sites may be challenging because this study was conducted over the course of six consecutive months. This is one of the limitations of this study.

## Conclusions and recommendations

NGU trap was found to be the most effective trap for catching *G. pallidipes* in all vegetation types except in riverine forests in both parks. It was also observed that the mean fly catch was

significantly higher in WGL areas than the other vegetation types. Nech Sar Park had the highest fly density when compared with the Maze National Park. This might be because of the control activities that are actively underway in and around the Maze Park, which is a good indicator to sustainably strengthen these activities until tsetse elimination is achieved in the area. *G. pallidipes* was the predominant species. Finally, NGU traps could be used in tsetse surveillance, monitoring and evaluation of the impact of control programs implemented in the region. This means one can easily choose NGU trap in *G. pallidipes* monitoring and control activities in the region.

## Acknowledgments

The authors would like to thank Dr. Zerihun W/Senbet who contributed a lot providing technical and professional assistances. The greatest contribution of Mr. Aschenaki Kalissa, the head of Tsetse and Trypanasomosis Control and Investigation Centre, to facilitate the necessary field materials for field work is also highly appreciated.

## Author Contributions

**Conceptualization:** Netsanet Asfaw, Fekadu Massebo.

**Data curation:** Netsanet Asfaw, Fekadu Massebo.

**Formal analysis:** Netsanet Asfaw, Fekadu Massebo.

**Investigation:** Netsanet Asfaw, Fekadu Massebo.

**Methodology:** Netsanet Asfaw, Fekadu Massebo.

**Project administration:** Fekadu Massebo.

**Resources:** Netsanet Asfaw, Fekadu Massebo.

**Supervision:** Berhanu Hiruy, Netsanet Worku, Fekadu Massebo.

**Validation:** Netsanet Asfaw, Berhanu Hiruy, Netsanet Worku, Fekadu Massebo.

**Visualization:** Netsanet Asfaw, Berhanu Hiruy, Netsanet Worku, Fekadu Massebo.

**Writing – original draft:** Netsanet Asfaw, Netsanet Worku, Fekadu Massebo.

**Writing – review & editing:** Netsanet Asfaw, Berhanu Hiruy, Netsanet Worku, Fekadu Massebo.

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
