## [Decision Letter · Decision Letter 0]

1 Sep 2022

Dear Mr. Asfaw,

Thank you very much for submitting your manuscript "Evaluating the Efficacy of Various Traps in Catching Tsetse Flies at Nech Sar and Maze National Parks, Southwestern Ethiopia : An Implication for  Trypanosoma Vector Control" for consideration at PLOS Neglected Tropical Diseases. As with all papers reviewed by the journal, your manuscript was reviewed by members of the editorial board and by several independent reviewers. In light of the reviews (below this email), we would like to invite the resubmission of a significantly-revised version that takes into account the reviewers' comments. 

Thank you for your submission to PLoS NTDs. The reviewers felt that your manuscript was of interest, however a primary issue with the manuscript in its current form is that the choice of statistical tools utilized for the analysis was inappropriate given the data collected. The analysis should be performed using a non-parametric test as the data collected is count based. The reviewers also indicate that the manuscript requires additional proof reading to identify and correct grammatical issues. Please read through the reviewers comments and address the points raised in a revised version of the manuscript. We look forward to receiving a an updated version for evaluation.

We cannot make any decision about publication until we have seen the revised manuscript and your response to the reviewers' comments. Your revised manuscript is also likely to be sent to reviewers for further evaluation.

Sincerely,

Geoffrey M. Attardo

Academic Editor

Epco Hasker

Section Editor

Thank you for your submission to PLoS NTDs. The reviewers felt that your manuscript was of interest, however a primary issue with the manuscript in its current form is that the choice of statistical tools utilized for the analysis was inappropriate given the data collected. The analysis should be performed using a non-parametric test as the data collected is count based. The reviewers also indicate that the manuscript requires additional proof reading to identify and correct grammatical issues. Please read through the reviewers comments and address the points raised in a revised version of the manuscript. We look forward to receiving a an updated version for evaluation.

Reviewer's Responses to Questions

**Key Review Criteria Required for Acceptance?**

**Methods**

-Are the objectives of the study clearly articulated with a clear testable hypothesis stated?

-Is the study design appropriate to address the stated objectives?

-Is the population clearly described and appropriate for the hypothesis being tested?

-Is the sample size sufficient to ensure adequate power to address the hypothesis being tested?

-Were correct statistical analysis used to support conclusions?

-Are there concerns about ethical or regulatory requirements being met?

Reviewer #1: The objectives are clearly articulated and the design (Latin Square) is appropriate for assessment of field responses of tsetse the traps. I am however concerned about the statistical approach employed in analysis of the data. Given that the data was discrete/count of flies (non-continuous), ANOVA is not the appropriate tool for the analysis, unless the data was transformed. Non-parametric methods would have been most appropriate.

Reviewer #2: -The objectives of the study are clearly articulated and the study design is appropriate

-The statistical analyses used were not appropriate for the type of data collected. Tsetse catches are counts and thus can not be analyzed using General Linear Models like ANOVA (which are suitable for continuous variables) without being transformed. Generalized Linear Models are more suitable for counts.

Reviewer #3: The paper is clear and very well written, and gives insight into the efficiency of different types of traps in catching tsetse flies at different areas and vegetation types. This evaluation is important because will allow to determine the real impact of control programs implemented in the region. The materials and methods are clearly indicated.

**Results**

-Does the analysis presented match the analysis plan?

-Are the results clearly and completely presented?

-Are the figures (Tables, Images) of sufficient quality for clarity?

Reviewer #1: The results are well written and presented.

Reviewer #2: Due to the flawed statistical analyses indicated above, the results are unreliable.

Reviewer #3: The results reflect to the methodology are clearly presented.

**Conclusions**

-Are the conclusions supported by the data presented?

-Are the limitations of analysis clearly described?

-Do the authors discuss how these data can be helpful to advance our understanding of the topic under study?

-Is public health relevance addressed?

Reviewer #1: The conclusions are appropriate and are supported by the data presented. However, the data might need to be analysed again using appropriate tool.

Reviewer #2: The conclusions could be erroneous as the results are unreliable due to the flawed statistical analysis

Reviewer #3: The results are also adequately discussed.

**Editorial and Data Presentation Modifications?**

Reviewer #1: The results are well written and presented.

Reviewer #2: (No Response)

Reviewer #3: Minor revision

**Summary and General Comments**

Reviewer #1: The MS by Asfaw et al., present a very clear and straightforward comparative assessment of efficacy of the three different traps (Ngu, Biconical and sticky) in four ecological zones (wood grassland, bush land, forest, and riverine forest) using tried, tested and approved methods. The findings are well written and presented. I am however concerned about two major issues. First, the statistical tool used for analysis of the data (ANOVA) was inappropriately applied or should have been substituted by an non-parametric method suitable for the count data that was analysed. The second concern is the lack of novelty in the findings, given that the results only appears to confirm other results, what is already known about relative performance of these devices as captured by the authors in the discussions and elsewhere. G. pallidipes is a savannah tsetse species while G. fuscipes is riverine species, information which is well documented, and which the authors appear to confirm. Other minor concerns is that the " personal communication" in line 254 should be qualified.

Reviewer #2: The manuscript requires substantial improvement in gramma and the authors should consult a statistician for their data analysis.

Reviewer #3: This is a technical report about “Evaluating the Efficacy of Various Traps in Catching Tsetse Flies at Nech Sar and Maze National Parks, Southwestern Ethiopia : An Implication for Trypanosoma Vector Control”. 

The paper is clear and very well written, and gives insight into the efficiency of different types of traps in catching tsetse flies at different areas and vegetation types. This evaluation is important because will allow to determine the real impact of control programs implemented in the region. The materials and methods are clearly indicated and the results reflect this. The results are also adequately discussed.

It is recommended that the manuscript be accepted for publication.

Some minor errors and modifications are suggested here below.

Abstract

Line 34: remove “s” to traps in “…skiny traps”

Author summary

Line 46: add “African” before “Animal trypanosomosis”

Line 46: add “agents” after “Animal trypanosomosis (Nagana) and Human African Trypanosomiasis (HAT)…”

Line 47: add “In animals,” before “…The disease is arguably the main constraint, …”

Line 50: “to control the disease" which disease (HAT or AAT)? Please precise 

Introduction

Line 76: remove “s” at the end of “SITs”

Line 86: put “,” after “In Ethiopia”

Materials and methods

Line 109: change “0” by “°” “…200C and 340C….”

Figure 1: Kulfo and Maze rivers must be displayed along their lengths on the map

Line 126-127: the 12 traps were deployed in each study site or each vegetation? Please clearify the sentence.

Lines 127-128: change “In the six consecutive months, a total of 72 traps per study site were deployed.” to “In the six consecutive months, a total of 72 traps were deployed per study site.”

Lines 129-130: the sentences “The type of trap used at the first trapping point in each type of vegetation was chosen at random.” has been repeated, please delete.

Line 137: for six months collection, the authors indicated that this makes possible to observe the seasonal variation in fly density. 

I am not sure that this is enough to observe the seasonal variation. This will be possible if the experiments were conducted over the 12 months.

Data management and analysis 

1. Authors must first test the normality of the data before comparative analyses.

2. Authors need to submit the data analysis details in Rmarkdown file to make the follow up of the data is clear and simple.

Results

Line 166: It is not possible to start a sentence with numbers, please replace write “82.7%” in letters first 

Line 166: harmonize the writing of "park" in the manuscript; either "Park" or "park"

Line 170-173: these results are specific to Glossina pallidipes, please specify, since the few G. fuscipes have been collected using sticky traps

Line 174: remove “(FTD)” in the table and replace the table tittle by “Mean density of tsetse flies (Fly per tap per day) in Nech Sar and Maze National Parks, southwest Ethiopia”

Line 175: remove “FTD = Fly per tap per day”

Line 183: add standard deviations to histograms (Figure 2)

Line 185: remove “s” to “traps”

Line 188: add “s” to “type” after vegetation

Line 201: add “except in riverine forest” after “vegetation types…”

Lines 208-210: Table 3 presents results of multiple comparisons, which was not specified in the "data analysis" section

Tables 3 and 4: 

1. for the column “Mean ± SE”, harmonize the number of digits after the decimal point

2. what is the unit of “Mean ± SE”??

Discussion

Delete from line 219 to line 224 because it is a repetition of the results

Line 244: change “0” by “°” “…200C and 340C….” 

References

Write species names in italics

PLOS authors have the option to publish the peer review history of their article (what does this mean?). If published, this will include your full peer review and any attached files.

Reviewer #1: Yes: Dr. Paul Mireji

Reviewer #2: No

Reviewer #3: No
---

## [Decision Letter · Decision Letter 1]

22 Nov 2022

Dear Mr. Asfaw,

Thank you very much for submitting your manuscript "Evaluating the Efficacy of Various Traps in Catching Tsetse Flies atNech Sar and Maze National Parks, Southwestern Ethiopia: An Implication for  Trypanosoma Vector Control" for consideration at PLOS Neglected Tropical Diseases. As with all papers reviewed by the journal, your manuscript was reviewed by members of the editorial board and by several independent reviewers. The reviewers appreciated the attention to an important topic. Based on the reviews, we are likely to accept this manuscript for publication, providing that you modify the manuscript according to the review recommendations. 

Sincerely,

Epco Hasker

Section Editor

Epco Hasker

Section Editor

Reviewer's Responses to Questions

**Key Review Criteria Required for Acceptance?**

**Methods**

-Are the objectives of the study clearly articulated with a clear testable hypothesis stated?

-Is the study design appropriate to address the stated objectives?

-Is the population clearly described and appropriate for the hypothesis being tested?

-Is the sample size sufficient to ensure adequate power to address the hypothesis being tested?

-Were correct statistical analysis used to support conclusions?

-Are there concerns about ethical or regulatory requirements being met?

Reviewer #2: -The objectives of the study are clearly articulated and the study design is appropriate. 

-The statistical analysis supports the conclusion.

Reviewer #3: (No Response)

**Results**

-Does the analysis presented match the analysis plan?

-Are the results clearly and completely presented?

-Are the figures (Tables, Images) of sufficient quality for clarity?

Reviewer #2: The analysis presented matched the analysis plan and the results are clearly presented.

Reviewer #3: (No Response)

**Conclusions**

-Are the conclusions supported by the data presented?

-Are the limitations of analysis clearly described?

-Do the authors discuss how these data can be helpful to advance our understanding of the topic under study?

-Is public health relevance addressed?

Reviewer #2: The conclusions are supported by the data presented.

Reviewer #3: (No Response)

**Editorial and Data Presentation Modifications?**

Reviewer #2: (No Response)

Reviewer #3: (No Response)

**Summary and General Comments**

Reviewer #2: In their manuscript, Asfaw et al. evaluate the efficacy of the NGU, Biconical and Sticky traps against different species of tsetse in different vegetation types. The contents of manuscript are of importance for tsetse vector control in Ethiopia. However, the manuscript requires to address the following issues before it can be published. 

Introduction

Line 91: remove the word “big”. The authors should also consider giving a summary of the tsetse eradication program stating the control and monitoring methods being used. 

Line 98: sentence starting with “It will….” should be in past tense.

Materials and methods

Line 102: consider rewriting the first sentence to “Permission to conduct the study in the National Parks was obtained from xxxxx(Park Authorities in Ethiopia).

Lines 127 to 137: the authors should clearly indicate the number of Latin Square replicates per month per site in each vegetation. This could be one of the limitations of the study and should be highlighted in the discussion. The authors should also describe the traps in more detail. For example what were dimensions of the stick trap and what material was used to make the trap stick?

Lines 138 and 143: the authors should consider combining the subheadings “Tsetse fly collection” and “Tsetse fly species identification” into one subheading “Tsetse collection and species identification”. 

Line 139: should be “baited with urine”. What was the source of the urine? Was the urine fresh? How was the urine dispensed?

Lines 155 to 162: the authors should indicate how the calculated the means (lines 176 to 177) and fly densities (185 to 186) reported in the results section.

Results

Lines 166 to 170: the authors should consider indicating the actual numbers of each tsetse species collected by trap.

Line 169: write 82.7% in words

Discussion

The authors should discuss the limitation of the study.

Reviewer #3: (No Response)

PLOS authors have the option to publish the peer review history of their article (what does this mean?). If published, this will include your full peer review and any attached files.

Reviewer #2: No

Reviewer #3: No

Figure Files:

Data Requirements:

Reproducibility:

References

---

## [Editor Report · Decision Letter 2]

29 Nov 2022

Dear Mr. Asfaw,

We are pleased to inform you that your manuscript 'Evaluating the Efficacy of Various Traps in Catching Tsetse Flies at Nech Sar and Maze National Parks, Southwestern Ethiopia: An Implication for  Trypanosoma Vector Control' has been provisionally accepted for publication in PLOS Neglected Tropical Diseases.

Best regards,

Epco Hasker

Section Editor

Epco Hasker

Section Editor

---

## [Editor Report · Acceptance letter]

19 Dec 2022

Dear Mr. Asfaw,

We are delighted to inform you that your manuscript, "Evaluating the Efficacy of Various Traps in Catching Tsetse Flies at Nech Sar and Maze National Parks, Southwestern Ethiopia: An Implication for  Trypanosoma Vector Control," has been formally accepted for publication in PLOS Neglected Tropical Diseases.

Best regards,

Shaden Kamhawi

co-Editor-in-Chief

Paul Brindley

co-Editor-in-Chief
